# Synthesis and Antimicrobial Activity of 3D Micro–Nanostructured Diatom Biosilica Coated by Epitaxially Growing Ag-AgCl Hybrid Nanoparticles

**DOI:** 10.3390/biomimetics9010005

**Published:** 2023-12-23

**Authors:** Zhanar Bekissanova, Viorica Railean, Izabela Wojtczak, Weronika Brzozowska, Grzegorz Trykowski, Alyiya Ospanova, Myroslav Sprynskyy

**Affiliations:** 1Faculty of Chemistry and Chemical Technology, Al-Farabi Kazakh National University, Almaty 050040, Kazakhstan; zhanar.bekissanova@kaznu.edu.kz (Z.B.); aliya4@kaznu.kz (A.O.); 2Center of Physical-Chemical Methods of Research and Analysis, Almaty 050012, Kazakhstan; 3Department of Infectious, Invasive Diseases and Veterinary Administration, Institute of Veterinary Medicine, Nicolaus Copernicus University in Torun, Gagarina 7, 87-100 Torun, Poland; vioricarai@umk.pl; 4Interdisciplinary Center for Modern Technologies, Nicolaus Copernicus University in Torun, Wilenska 4, 87-100 Torun, Poland; 5Department of Environmental Chemistry and Bioanalytics, Faculty of Chemistry, Nicolaus Copernicus University in Torun, 7 Gagarina Str., 87-100 Torun, Poland; izabelawojtczak1991@gmail.com; 6Division of Surface Science, Faculty of Chemical Technology and Engineering, Bydgoszcz University of Science and Technology, Kaliskiego 7, 85-796 Bydgoszcz, Poland; weronika.brzozowska@pbs.edu.pl; 7Department of Materials Chemistry, Adsorption and Catalysis, Faculty of Chemistry, Nicolaus Copernicus University in Torun, Gagarina 7, 87-100 Torun, Poland; tryki@umk.pl

**Keywords:** diatom biosilica, nanoparticles, Ag-AgCl hybrid nanoparticles, epitaxial growth nanoparticles, antibacterial activity

## Abstract

The 3D (three-dimensional) micro–nanostructured diatom biosilica obtained from cultivated diatoms was used as a support to immobilize epitaxially growing AgCl-Ag hybrid nanoparticles ((Ag-AgCl)NPs) for the synthesis of nanocomposites with antimicrobial properties. The prepared composites that contained epitaxially grown (Ag-AgCl)NPs were investigated in terms of their morphological and structural characteristics, elemental and mineral composition, crystalline forms, zeta potential, and photoluminescence properties using a variety of instrumental methods including SEM (scanning electron microscopy), TEM (transmission electron microscopy), EDX (energy-dispersive X-ray spectroscopy), XRD (X-ray powder diffraction), zeta-potential measurement, and photoluminescence spectroscopy. The content of (AgCl-Ag)NPs in the hybrid composites amounted to 4.6 mg/g and 8.4 mg/g with AgClNPs/AgNPs ratios as a percentage of 86/14 and 51/49, respectively. Hybrid nanoparticles were evenly dispersed with a dominant size of 5 to 25 nm in composite with an amount of 8.4 mg/g of silver. The average size of the nanoparticles was 7.5 nm; also, there were nanoparticles with a size of 1–2 nm and particles that were 20–40 nm. The synthesis of (Ag-AgCl)NPs and their potential mechanism were studied. The MIC (the minimum inhibitory concentration method) approach was used to investigate the antimicrobial activity against microorganisms Klebsiella pneumoniae, Escherichia coli, and Staphylococcus aureus. The nanocomposites containing (Ag-AgCl)NPs and natural diatom biosilica showed resistance to bacterial strains from the American Type Cultures Collection and clinical isolates (diabetic foot infection and wound isolates).

## 1. Introduction

Recently, diatom biosilica has been used to synthesize new 3D microstructured composite materials. The basic 3D structure of diatom biosilica, which plays the role of a substrate in gold modification, can be used as a Surface-Enhanced Raman-Scattering Support [1]. The authors of [2] synthesized three-dimensional (3D) diatom@TiO_2_@MnO_2_ composites for supercapacitors or through the metabolic doping of titanium on diatomaceous biosilica to improve optical properties [3]. Diatom biosilica’s composition and structure can be altered, which could increase the number of potential applications by introducing new functions and active components. The diversity of the diatom silica exoskeletons morphology (over 100,000 different species) gives scientists the opportunity to choose the diatom species that can be easily modified and correspond to the desired application, for example, in biomedicine [4,5], in drug delivery systems for anticancer therapeutics [6,7,8,9], for bone tissue engineering [10], as a catalyst [8] and photocatalysts [11] also suitable for semiconductor nanolithography and spectroscopy, for photoluminescence [12,13], in bio-triboelectric generators [14], and in biosensing application [15,16]. The diatom exoskeletons (called frustules) consist of siliceous walls perfectly decorated by a spatially periodic porous framework, creating unique openwork three-dimensional silica structures for nano–microscale organization. The periodicity, as well as the morphology and size, of these pores are strictly dependent on the species of diatom and facilitates the potential use of diatom frustules as natural microlenses, optical filters, and laser resonators [17,18]. Various diatom species such as *Halamphora subturgida* [19], *Thalassiosira pseudonana* [9,20], and *Amphora subtropica* [7] can act as a source of biosilica. An eco-friendly source of silica is diatoms, which can be actively cultivated in laboratories and photobioreactors [21]. Biomedical applications require intensive cleaning of diatom biosilica to produce safe and biocompatible material [4]. For this reason, biosilica and biosilica-based nanomaterials have great potential as eco-friendly functional materials in microsystem and bionanotechnology manufacturing worldwide [22].

The use of nanosilver as a biocidal material was registered more than a hundred years ago [23]. Mathew Carey Lea, for the first time in 1889, reported the synthesis of silver colloids stabilized with sodium citrate [24]. Currently, the main existing routes for the synthesis of silver nanoparticles are physical, chemical, and biological methods [25]. The widely used method for the chemical synthesis of metallic silver nanoparticles (AgNPs) is the wet reduction of silver ions to a colloidal state in the presence of various reducing agents [26,27,28]. Silver chloride nanoparticles (AgClNPs) can be obtained through an ion-exchange reaction by introducing salts NaCl [26,29] or AlCl_3_•6H_2_O [30]. There are few reports on the synthesis of hybrid silver nanoparticles by combining metallic silver and silver chloride nanoparticles. The synthesized hybrid nanocomposites CEP-Ag/AgCl/ZnO, where pectin played the role of matrix and stabilizing agent, showed excellent photocatalytic and antibacterial activity [29]. The prepared cellulose/Ag/AgCl hybrid nanocomposites also showed antibacterial activity [30]. The authors of [31] reported on synthesized hybrid nanocomposites of AgCl nanocrystals and metallic silver nanowires. The introduction of Ag/AgCl nanoparticles into cancer cells showed a therapeutic effect and a minor cytotoxic effect [32]. Pure AgNPs and AgClNPs cannot show the same results as Ag/AgCl nanoparticles. AgCl-Ag hybrid particles have a lesser cytotoxic effect compared to pure AgCl nanoparticles [32], and low cytotoxicity was also detected in alginate-based films containing Ag/AgCl nanoparticles [33]. Potential antibacterial agents of the composition MOFs of Cu(I)bpyCl (bpy = 4,4′-bipyridine) formulated with AgCl/Ag nanoparticles with possible synergistic effects were synthesized [34]. The composites based on AgCl/Ag nanoparticles coated on reduced graphene oxide have antimicrobial potential and the action of healing burn wounds [35]. In one study, AgCl/Ag nanoparticles decorated layered double hydroxide, which plays the role of a support and prevents the aggregation of nanoparticles [36]. Natural matrices have the necessary properties for the synthesis of biocompatible non-toxic composite materials [37,38]. When composite materials based on natural matrices are synthesized, they prevent the emission of silver nanoparticles in the environment and the aggregation of silver nanoparticles [36]. The above shows that the synthesis of hybrid composites containing AgCl/Ag nanoparticles can find a wide application.

The aim of this research is to synthesize a new 3D nano–microstructured composite with antimicrobial activity based on diatom biosilica coated by epitaxially growing Ag-AgCl hybrid nanoparticles. The antimicrobial activity was investigated against Gram-positive bacteria *Staphylococcus aureus* (from the American Type Culture Collection and diabetic foot infection isolates) and Gram-negative bacteria *Klebsiella pneumoniae* and *Escherichia coli* strains (from the American Type Culture Collection and wound isolates).

## 2. Materials and Methods

### 2.1. The Diatom Biosilica

Diatom biosilica was obtained from diatom microalgae *Pseudostaurosira trainorii* (the Culture Collection of Baltic Algae, Institute of Oceanography, University of Gdańsk, Poland). The characterization of the diatom biosilica structure, the physicochemical properties of biosilica, the diatom cultivation method, and the procedure for obtaining purified 3D-structured biosilica from diatoms are given in previous publications [39,40].

### 2.2. Instrumental Methods and Characterization

#### 2.2.1. Elemental Composition Analysis

The chemical composition of hybrid composite (Ag-AgCl)NPs/biosilica was characterized using scanning electron microscopy (LEO 1430 VP, Electron Microscopy Ltd., Cambridge, UK) coupled with energy-dispersive X-ray (detector XFlash 4010, Bruker AXS, Bremen, Germany).

#### 2.2.2. X-ray Diffraction Analysis

The mineral composition of 3D micro–nanostructured hybrid composite (Ag-AgCl)NPs/diatom biosilica was characterized with the XRD method using a Philips X ‘Pert Pro diffractometer (Malvern Pananaliytical Ltd., Malvern, UK) with Cu-Kα-radiation (γ = 0.1541 nm, 40 kV, 30 mA). X-ray diffraction data at 0.01 step sizes was recorded in the angle range of 5–100° 2θ.

#### 2.2.3. Transmission Electron Microscopy Studies

The determination of sizes and distribution of nanoparticles in prepared hybrid composite was carried out using transmission electron microscopy (FEI Tecnai F20 X-Twintool, FEI Europe, Frankfurt/Main, Germany).

#### 2.2.4. Zeta-Potential Measurements

Using the Zetasizer Nano Series (Malvern Instruments, Malvern, UK), the zeta-potential values of synthesized composites containing 4.61 and 8.49 silver were determined. The tested samples (at a concentration of 0.05 mg/mL) prior to measurement were sonicated for 360 min at 25 °C in a Polsonic ultrasonic bath with water at a predefined pH range (from 2 to 12). For each sample, there were three replicate measurements.

#### 2.2.5. Photoluminescence Analysis

The photoluminescent properties of the 3D micro–nanostructured composite were evaluated using the Hitachi F-2500 Fluorescent spectrophotometer (Tokyo, Japan) with xenon lamp at three different excitation wavelengths at approximately 20 °C. The analysis was performed on solid composites which were placed in a specialized cuvette. The range of measurements was 285–800 nm, in accordance with the functions of the instrument. The analyzed wavelength ranges for excitation wavelengths 270 and 420 nm, respectively, were used to describe the photoluminescence properties.

### 2.3. Synthesis of 3D Micro–Nanostructured Composite

The impregnation method was used for obtaining 3D micro–nanostructured hybrid (Ag-AgCl)NPs/diatom biosilica composites. Diatom biosilica was impregnated with aqueous AgNO_3_ solutions in two concentrations of Ag ions: 50 mg/L and 100 mg/L or 5 and 10% according to the weight of used diatom biosilica. The diatom biosilica mass of 100 mg was added to a AgNO_3_ solution (50 mL) with a specific concentration of silver ions. AgClNPs were formed with an exchange reaction where AgNO_3_ reacts with NaCl.
AgNO_3_(aq) + NaCl(s) → NaNO_3_(aq) + AgCl(s)(1)

The prepared suspension was stirred for 30 min, and then 0.1 M NaOH was added to basify it to pH = 9.
2AgNO_3_(aq) + 2NaOH(aq) → 2NaNO_3_(aq) + Ag_2_O(s) + H_2_O(2)

The precipitated silver oxide (Ag_2_O) was reduced using hydrogen peroxide. The reducing agent peroxide was used in a 1:3 molar ratio of AgNO_3_/H_2_O_2_. After that, the suspension was stirred at 300 rpm for 15 min for a total reduction of silver:Ag_2_O(s) + H_2_O_2_(aq) → 2Ag(s) + H_2_O(aq) + O_2_(g)(3)

The 3D micro–nanostructured hybrid (Ag-AgCl)NPs/diatom biosilica composite was centrifugated at 9000 rpm, the washing process was repeated five times with deionized water, and then the composite was dried at 110 °C for 12 h.

### 2.4. Antimicrobial Activity Study of the 3D Micro–Nanostructured Composite

The 3D micro–nanostructured hybrid (Ag-AgCl)NPs/diatom biosilica composite (4.61% Ag/biosilica, 8.49% Ag/biosilica) was evaluated against *Staphylococcus aureus* ATCC 700699, *Klebsiella pneumonia* ATCC 10031, *Escherichia coli* ATCC 10031) purchased from American Type Culture Collection (ATCC) while pathogenic isolates *Staphylococcus aureus* ATCC 33591 THL (DFI), *Klebsiella pneumoniae* 9295_1 CHB (WI), *Escherichia coli* MB 11464 1 CHB (WI) were obtained from the Centre for Modern Interdisciplinary Technologies collection. For this process, solutions of (Ag-AgCl)NPs/diatom biosilica composite were prepared at the following concentrations of 10 mg/mL, 5 mg/mL, 2.5 mg/mL, and 1.25 mg/mL. The experiment was performed according to the protocol suggested by the Clinical and Laboratory Standards Institute (CLSI) guidelines. The Clinical and Laboratory Standards Institute (CLSI) proposed a protocol for the experiment. Details are described by our group in Z. Kubasheva et al., 2020 [41]. The resazurin method was used according to the production technology to display the minimum inhibitory concentration (MIC). The natural silver-free diatom biosilica substrate was used as a control. The experiment was carried out three times.

## 3. Results

### 3.1. Characterization of the 3D Micro–Nanostructured Diatom Biosilica

The morphology and structural features of the 3D diatom biosilica that were observed on the scanning electron microphotographs of the diatoms, as well as the cultivation setup and colonial chains of the living diatoms obtained with light microscopy, are demonstrated in Figure 1.

The scanning electron microphotographs (Figure 1C,D) show that the cleaned diatom biosilica (siliceous skeleton called a frustule) of the cultivated diatom species is exhibited in a “microsaucer” shape with a diameter of 4–5 μm and has a specific hierarchical 3D micro–nanostructure. The silica walls of the diatom frustules are decorated with unique patterns with periodic parallel lines of oval pores ranging in size from 100 to 200 nm in diameter. According to previously obtained test results on low-temperature nitrogen adsorption and desorption isotherms, the authors of [39] tested biosilica that also possessed micropores with a size of about 1–1.5 nm and straight-slit mesopores in the size range of 10–20 nm. The presence of straight-slit mesopores in the biosilica was indicated by a hysteresis loop of the H4 type in the obtained nitrogen adsorption and desorption isotherms. The BET surface area of the diatom biosilica was calculated to be 16.9 m^2^/g. The obtained XRD pattern for the diatom biosilica was characteristic of an amorphous hydrated silica of the opal-A type. Opal-A was also indicated by the low atomic ratio of oxygen to silicon (1.9) that was detected using the TEM–EDX analysis [39].

The results of FTIR spectroscopy analysis [39,40] showed the presence of intense absorption bands centered at 1062 cm^−1^, 943 cm^−1^, 797 cm^−1^, and 447 cm^−1^ and broadband at around 3650–3000 cm^−1^ that are assigned to the characteristic structural bonds (Si–O–Si, Si–O–H, Si–O, H–O–H) in the amorphous hydrated silica. The thermogravimetric analysis [31] of the diatom biosilica demonstrates three main, distinct stages of weight loss during heating, which may be attributed to the loss of loosely held physically bonded water (5.5%, 30–125 °C, dehydration process) to the release of water as the condensation product of the lost vicinal silanol groups (5%, 250–675 °C 6%, dehydroxylation process) and to the loss of internal isolated hydroxyl groups (1.3%, 675–900 °C, dehydroxylation process) from the biosilica structure.

### 3.2. Energy-Dispersive X-ray Spectroscopy (EDX) Studies

SEM–EDX spectra and the SEM image of the raw diatom biosilica and biosilica decorated with silver are presented in Figure 2. We can observe on the SEM image and SEM mapping that decorated silver was uniformly dispersed on the matrix of the natural diatom biosilica. The raw biosilica consisted of mainly oxygen and silicon. In the prepared composites, the source of the silver ions was the silver nitrate solutions that were used in two concentrations (50 mg/L and 100 mg/L), and the silver content was equal to 4.61%, and 8.49%, respectively.

The results of the molar and mass correlations of silver and chlorine in the obtained composites are demonstrated in Table 1. When the silver concentration of 50 mg/L was used in the synthesized solutions, the ratio of nanoparticles AgClNPs: AgNPs in the composites was in the ratio of 86:14 as a percentage. But, already, when the silver concentration increased to 100 mg/L, the relative content of AgCl nanoparticles significantly decreased, and the ratio of AgClNPs:AgNPs was equal to 51:49.

### 3.3. Powder X-ray Diffraction Results

The XRD patterns of the natural biosilica and (Ag-AgCl)NPs/diatom biosilica composites are demonstrated in Figure 3. XRD analysis revealed a broad peak in the range of 20° to 35°, confirming the presence of the amorphous nature of the biosilica. This diatom biosilica was discovered as hydrated similar to opal-A [39]. Both silver chlorides and reduced metallic silver nanoparticles were found in the produced composites, according to the X-ray spectra. The presence of nanoparticles of metallic silver is revealed by the appearance of three peaks with 2θ values of 38.46°, 64.85°, and 77.47° equivalent to the reflections of the crystallographic planes (111), (200), and (220), respectively (JCPDS No. 04-0783). The appearance of five diffraction peaks with 2θ values of 28.10°, 32.51°, 46.43°, 54.96°, and 57.85° with the respective crystallographic planes (111), (200), (220), (311), and (222) corresponding to the reflections points to the existence of silver chloride nanoparticles (JCPDS No. 31-1238).

### 3.4. Transmission Electron Microscopy Results

The determination of the sizes, shape, and distribution of the nanoparticles in the prepared hybrid composite was carried out using transmission electron microscopy. The micrographs of the synthesized (Ag-AgCl)NPs/diatom biosilica hybrid composites with a silver content of 8.49%, a histogram of the size distribution of the nanoparticles, and STEM–EDS line-scan profiles of the (AgCl-Ag)NPs with the Volmer–Weber epitaxial growth model are shown in Figure 4.

According to the obtained results, we proposed to form the nucleation of silver chloride on the diatom biosilica support in the first order and further the epitaxial growth of the metallic silver nanoparticles. The hybrid (Ag-AgCl)NPs have quasi-spherical forms that are homogeneously distributed on the diatom biosilica, according to the TEM images Figure 4A. In the TEM micrographs, the (Ag-AgCl)NPs hybrid nanoparticles can be observed in dark and light shades. Particles with darker shades are AgNPs, while lighter shades may be AgClNPs [42]. The dominant nanoparticles are from 5 to 15 nm Figure 4B. Nanoparticles of 1–2 nm and particles of 20–40 nm are also present Figure 4B. Moreover, Figure 4 exposes the presence of crystallites called parallel twins [43]. Synthesized nanoparticles can be categorized according to the twinning boundaries in Figure 4A as Axially Twined (AT) and Layer Twinned (LT) [44].

The elemental profile of two hybrid nanoparticles and the epitaxial growth model of the hybrid (Ag-AgCl)NPs are presented in Figure 4C. The cross-sectional profile and model data revealed an event known as epitaxy or epitaxial growth [45,46]. We proposed to form the nucleation of silver chloride on the support of the diatom biosilica carrier and further the epitaxial growth of the metallic silver nanoparticles (Figure 4C). Silver chloride plays the role of nucleation in connection with its formation in an early stage. Silver chloride nanoparticles are the first to be created on the biosilica surface as a result of the double-displacement reaction of silver nitrate with sodium chloride because silver chloride has very low solubility. The nucleation of the AgCl nanoparticle and further growth of metallic silver nanoparticles was observed in [44]. The authors of [47] revealed that the smaller AgCl nanocubes are a more potent heterogeneous nucleant for the growth of metallic silver nanowires. In one study, the formation was observed in the form of the heteroepitaxial growth of gold on the surface of silver chloride in order to obtain nanosized gold nanoparticles [48].

### 3.5. Zeta Potential

The dependence of the zeta-potential values (ζ-potential) on the pH for the biosilica and hybrid composite (AgCl-Ag) NPs/diatom biosilica are shown in Figure 5.

In the assumed pH range, the zeta-potential value ranges from +7.73 mV to −81.33 mV. It was shown that electrokinetic stability is strongly dependent on the type of material. The addition of silver to the diatom biosilica results in a decrease in the zeta potential, which is particularly evident for pH > 3. It can be observed that the isoelectric point (IEP) for the pure biosilica was not obtained in the given pH range. Data from the literature suggest that the reason for this is the polar measurement environment, which determines the existence of an IEP for biosilica around pH = 1.7 [49,50]. This is related to the negative surface charge, which may be due to the acidic behavior of the surface silanol groups [51]. The addition of silver to the diatom biosilica shifted the IEP values toward higher pH values. When the percentage of silver addition in the composite is higher, then the value of the IEP shift is larger. This is because of the change in the surface charge after the adsorption of hybrid silver nanoparticles, confirming the successful modification of the biosilica surface. Above pH = 10, a significant decrease in zeta-potential values is observed for all types of materials. This is likely due to the deprotonation of all hydroxyl groups on the diatom biosilica surface [52]. Moreover, in this pH region, hybrid silver nanoparticles show high stability due to significant electrostatic repulsions [53]. A significant increase in the negative value of the zeta potential of the diatom biosilica doped with silver nanoparticles compared to pure biosilica may be caused by an increase in the specific surface area of the biosilica due to the big surface area of the precipitated silver nanoparticles. A negative value for the zeta potential increases as a result of the increase in the number of hydroxyl groups adsorbed on the surface of the silver nanoparticles.

### 3.6. PL Spectroscopy

Figure 6 shows the photoluminescence spectra of the biosilica and prepared (Ag-AgCl)NPs/diatom biosilica composites containing different percentages of silver: 4.61% and 8.49%. Part A illustrates photoluminescence (FI) data obtained directly from the analysis, without any processing. Part B shows the data after normalization (NI) using OriginPro 9.1 software. The normalization process consisted of creating a baseline and subtracting the assigned values from the source data. Figure 6(A1,B1) present photoluminescence data in the emission wavelength range from 435 nm to 800 nm with an excitation wavelength of 420 nm. Figure 6(A2,B2) illustrate the PL spectra obtained for an excitation wavelength of 270 nm in the emission wavelength range from 285 nm to 530 nm.

According to the acquired photoluminescence spectra, four main types of photoluminescence activity (PL) can be identified in both composites. The initial type of PL is connected to excitation at 270 nm and emission at 335–425 nm in the UV spectrum. The strongest photoluminescence in this region was observed at 335, 380, 395, 397, 413, and 420 nm (Figure 6(B2)). The second type was associated with emission on the blue region of the visible spectrum (451–493 nm). For all varieties of composites, 420 nm excitation results in emission peaks at 450, 469, 482, and 493 nm (Figure 6(B2)). The green emission in the 508–524 nm range is a distinctive feature of the third type of photoluminescence activity. In this region of the visible-light spectrum, the most intense photoluminescence appears at 516 nm with 420 nm excitation. The fourth and final category of PL activity is connected to emissions in the red-light spectrum (630–780 nm). The peak with the highest intensity is at 635 nm and 420 nm excitation. With the same excitation wavelength, a small peak at 798 nm is also discernible.

Part A illustrates the photoluminescence (FI) data obtained directly from the analysis, without any processing. Part B shows the data after normalization. Figure 6(A1,B1) present photoluminescence data in the emission wavelength range from 435 nm to 800 nm with an excitation wavelength of 420 nm. Figure 6(A2,B2) illustrate PL spectra obtained for an excitation wavelength of 270 nm in the emission wavelength range from 285 nm to 530 nm.

The location and shape of the photoluminescence peak are almost independent of the excitation wavelength. From the spectra, it can be seen that there is a shift in the Ag characteristic bands toward higher energy as the size of the Ag nanoparticles in the composites decreases [54]. This shift can also be caused by the shape of the obtained nanoparticles [55]. A photoluminescence peak at 335 nm for an excitation wavelength of 270 nm is observed for Ag nanoparticles of small size (up to 10 nm). The luminescence band centered at 469 nm (for an excitation wavelength of 270 nm) is characteristic of Ag nanoparticles with a size around 25 nm [53], whose presence was confirmed with TEM analysis (Figure 4). Similar to the photoluminescence of noble metals, the emission in the blue-light region of the observed silver composites can be attributed to the radiative recombination of electrons at the Fermi level and holes in the sp or d band [56]. The chemical reduction of silver also causes a pronounced shift in the Ag characteristic peak toward red light, which may be the reason for the presence of a peak at an emission wavelength of 798 nm with 420 nm excitation. A detailed interpretation of the photoluminescence spectrum for diatomaceous biosilica was already described in the work of Sprynskyy et al. [39].

The distinct quenching of luminescence intensity may be related to charge transfer from the diatom biosilica to silver nanoparticles, which act as a plasmonic absorber of visible light (photons) that disturbs/slows photo-induced carrier recombination [57,58], which also explains the composites’ high photocatalytic effectiveness in terms of antibacterial activity. Photogenerated electron-hole pairs in such composites can be easily transported to the heterostructure interface, resulting in increased photocatalytic activity [57,59].

### 3.7. Antibacterial Activity

The synthesized hybrid composite (AgCl-Ag)NPs/diatom biosilica exhibited similar antimicrobial effects against all bacteria strains purchased from the American Type Culture Collection and clinical isolates (diabetic foot infection and wound isolates). The MIC value in both cases was found to be 1.25 mg/mL (Figure 7). It is noteworthy to specify the fact that the 4.61% (AgCl-Ag)NPs/biosilica containing lower silver content (0.058 mg/mL) displayed the same antimicrobial effect as the 8.49% (AgCl-Ag)NPs/biosilica where the silver content was determined to be higher. One interesting aspect needs to be specified: the raw diatom biosilica, used in the present research as a control, manifested antimicrobial activity showing an MIC value equal to 5 mg/mL for both species of *S. aureus:* ATCC 700699 and DFII isolates. In contrast, no inhibitory effect was noticed for *K. pneumonia* ATCC 10031 and WI isolate and *E. coli* WI isolate. However, the raw diatom biosilica exhibited an antimicrobial effect on the *E coli* from the ATCC 10031 collection with MIC value of 10 mg/mL. Composites with a predominant AgClNPs content versus AgNPs were found to have a greater inhibitory effect compared to AgNPs.

The results of previous studies show that AgClNPs possess substantial antibacterial and antifungal properties [58,60,61], and it was found that (Ag-AgCl)NPs demonstrate similar or stronger antimicrobial activity compared to AgNPs against Gram-negative and Gram-positive bacteria [60]. The higher antibacterial activity of AgClNPs against *E. coli* (MIC = 3.0 μg/mL) and agricultural pathogen *R. Solanacearum* (MIC = 2.0 μg/mL) compared to that of biosynthetic AgNPs was reported [62]. Nanomaterial Ag/AgCl/rGO showed minimum inhibitory concentration (MIC) values against *E. coli* and *S. aureus* of 2 and 4 mg L^−1^, respectively (in terms of the Ag element) [35]. The antibacterial activity of the composite with AgCl/Ag nanoparticles against *E. coli* showed the MIC of ~7.8 μg mL^–1^, while the MIC value was ~16 μg mL^–1^ against *S. aureus* [34].

The mechanism of action of silver nanoparticles leading to cell death has various hypotheses [63,64]. Scientists suggest that cell death is due to cell membrane destruction, enzyme initiation, the inhibition of the electron transfer chain (ETC), damage to nucleic acid and DNA, and oxidative stress released through reactive oxygen species (ROS) [65]. The mechanism of the antibacterial activity of the synthesized hybrid nanoparticles (AgCl-Ag)NPs in the present research approaches the photocatalytic hypothesis, but we do not have enough data. Previously, the synergistic effect of hybrid nanocolloids (AgCl/BAC nanocolloids) and hybrid nanocomposites (AgClNPs/AgNPs/diatomite) on microorganisms was reported [41,66,67]. The authors observed the synergistic photocatalytic antibacterial activity of composites Ag-AgCl/G-ZnFe_2_O_4_, and the antibacterial effect was significantly different depending on the lighting [68]. Our earlier research [41] also showed the slightly higher antibacterial activity of silver chloride nanoparticles. Moreover, silver chlorides are more stable and, therefore, have lower emissions of silver ions and, thus, lower cytotoxicity and greater environmental safety [60]. In our case, we also expected a synergistic photocatalytic effect in the antibacterial activity of the synthesized (Ag-AgCl)NPs hybrid nanocomposites based on biosilica. Indeed, the confirmation of such a hypothesis requires separate, more detailed, and precise tests, especially fluorospectrometric tests.

## 4. Conclusions

In summary, a new (Ag-AgCl)NPs/diatom biosilica composite containing epitaxially growth metallic silver nanoparticles over silver chloride nanoparticles was successfully synthesized. The characterization results showed that the prepared composites contained (Ag-AgCl)NPs hybrid nanoparticles mainly in the size range of 5–25 nm. The content of nanoparticles in the two synthesized composites was 4.6 mg/g and 8.4 mg/g with the AgClNPs/AgNPs ratio as a percentage of 86/14 and 51/49, respectively. Silver nanoparticles are often found in the quasi-spherical form of crystalline twins in Axially Twined (AT) and Layer Twinned (LT) with twinning boundaries.

The synthesized (Ag-AgCl)NPs/diatom biosilica composites exhibited high (1.25 mg/mL MIC of composite with silver amount 0.058 mg/mL) antibacterial activity against Gram-positive bacteria *Staphylococcus aureus* and Gram-negative bacteria *Klebsiella pneumoniae* and *Escherichia coli* originating from the American Type Culture Collection and clinical isolates (diabetic foot infection and wound isolates). Raw diatom biosilica also exhibited antibacterial activity. The synergistic effect of metallic silver nanoparticles, silver chloride nanoparticles, and diatom biosilica is possible.

The 3D micro–nanostructured diatom biosilica that was used as a natural support matrix for silver hybrid nanoparticles possesses high thermal and mechanical stability, is non-toxic and has good biocompatibility, and effectively prevents the aggregation of nanoparticles and their emission into the environment.

The results indicate that the synthesized (AgCl-Ag)NPs/diatom biosilica composites can be used as an effective antibacterial agent in various biomedical applications.

## Figures and Tables

**Figure 1 biomimetics-09-00005-f001:**
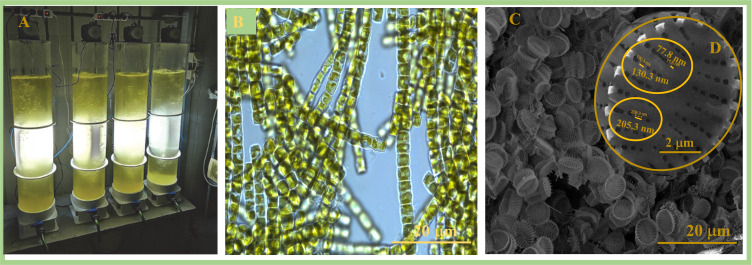
The diatoms cultivation setup (**A**), colonial chains of living diatoms (**B**), SEM images of a diatom frustule assembly (**C**), and single diatom frustule (**D**).

**Figure 2 biomimetics-09-00005-f002:**
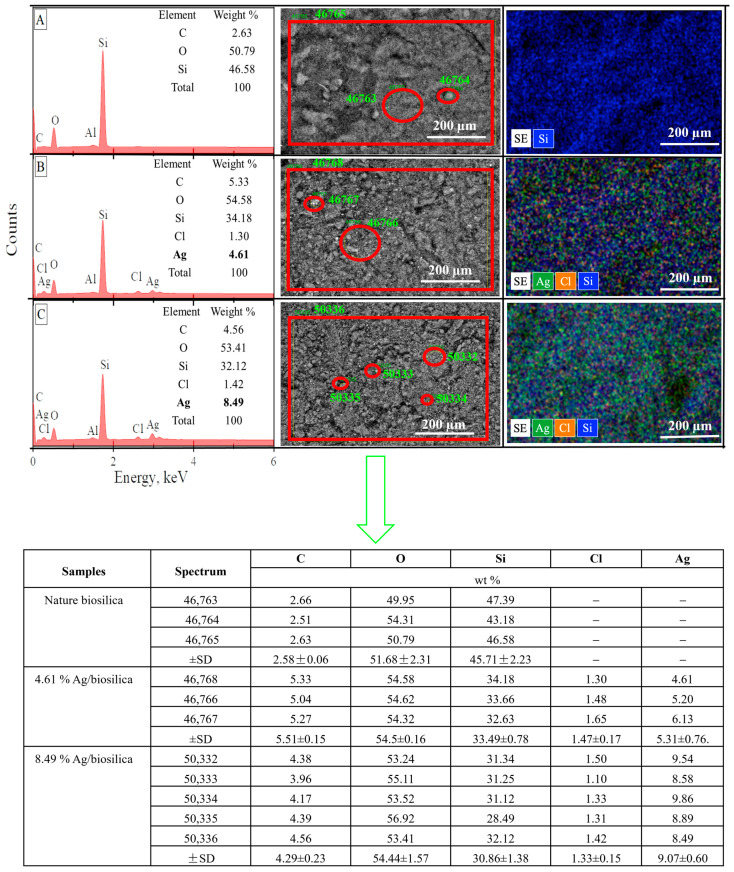
SEM–EDX spectra, SEM images, and mapping of diatom biosilica samples: (**A**) natural biosilica, (**B**,**C**) biosilica decorated with silver (4.61 and 8.49%, respectively).

**Figure 3 biomimetics-09-00005-f003:**
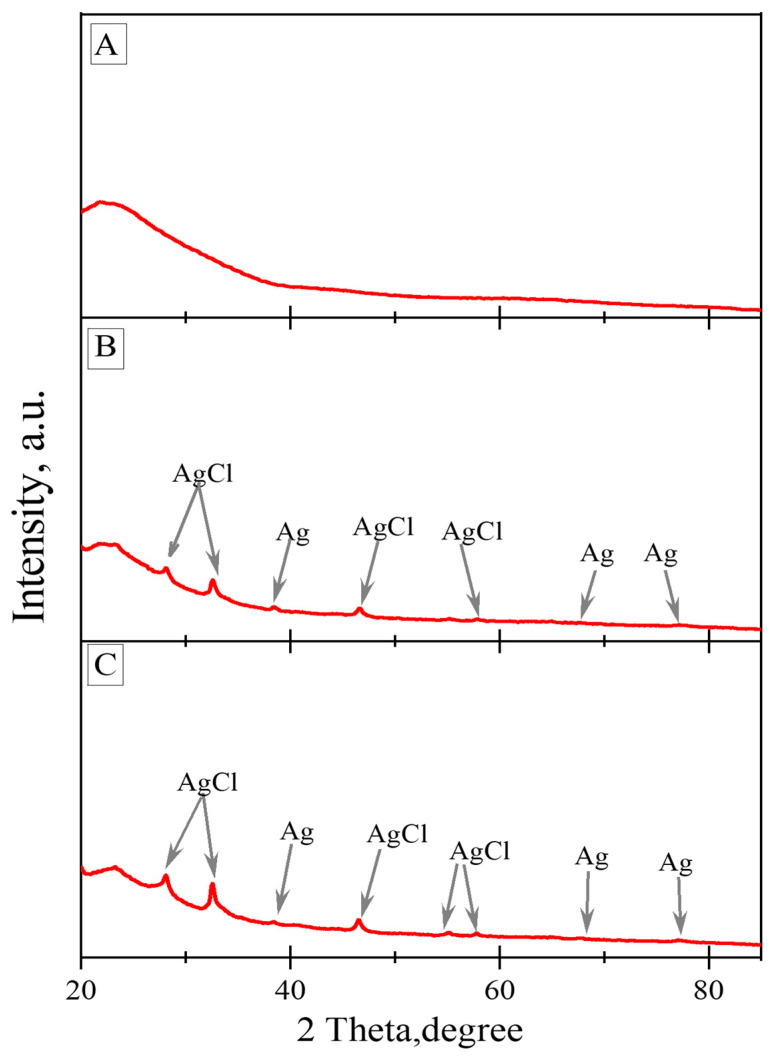
XRD patterns of diatom biosilica samples: (**A**) natural biosilica, (**B**,**C**) biosilica decorated with silver (4.61 and 8.49%, respectively); AgCl—silver chloride, Ag—metallic silver.

**Figure 4 biomimetics-09-00005-f004:**
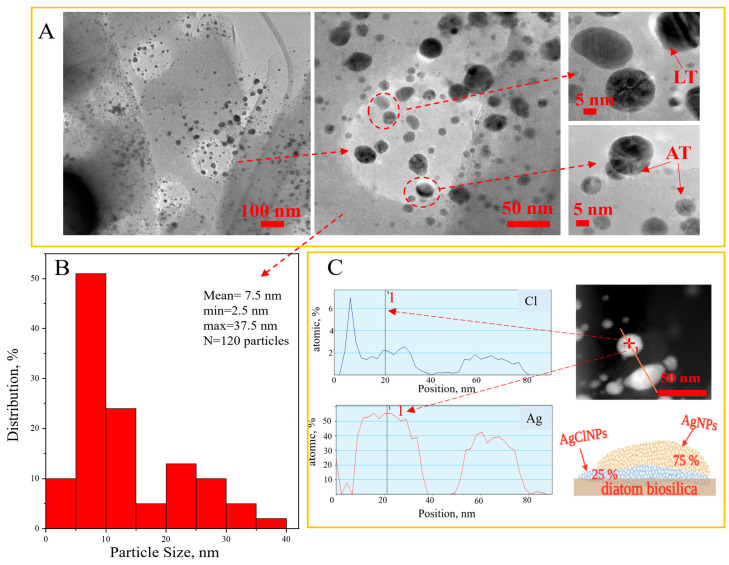
TEM micrographs of 3D micro–nanostructured hybrid (Ag-AgCl)NPs/diatom biosilica composite with silver concentration 8.49% (**A**); histogram of the size distribution of nanoparticles (**B**); and STEM–EDS line cross-sectional profiles of (Ag-AgCl)NPs with a model of hybrid nanoparticle structure according to the Volmer–Weber epitaxial growth model (**C**).

**Figure 5 biomimetics-09-00005-f005:**
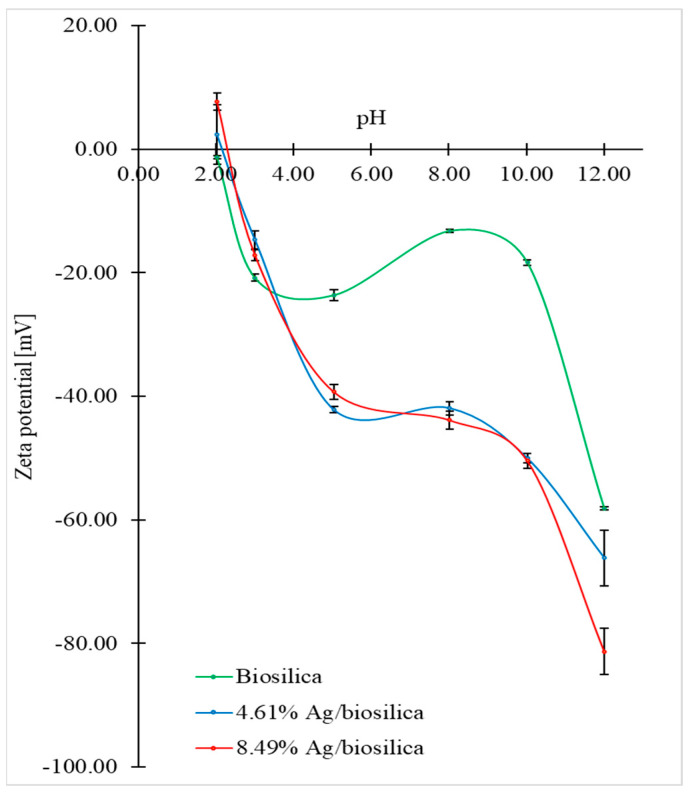
The dependence of zeta-potential value on pH for raw biosilica and hybrid nanocomposite (AgCl-Ag)NPs/diatom biosilica with 4.61% Ag and 8.49% Ag.

**Figure 6 biomimetics-09-00005-f006:**
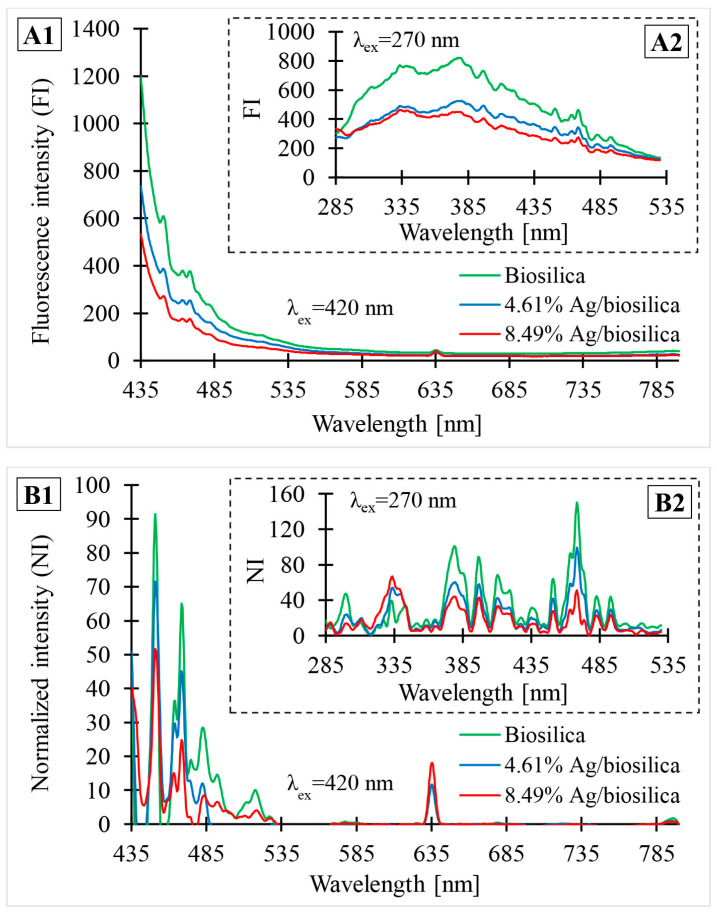
(**A1**,**A2**,**B1**,**B2**) Photoluminescence spectra of biosilica and composites containing different percentages of silver: 4.61% and 8.49%.

**Figure 7 biomimetics-09-00005-f007:**
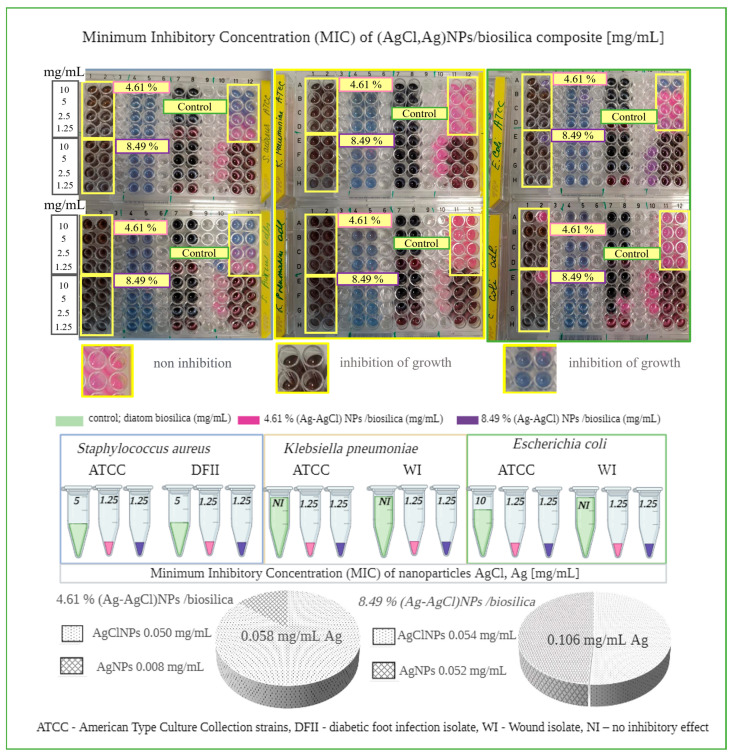
Antimicrobial activity assay and minimum inhibitory concentration of 3D micro–nanostructured hybrid (Ag-AgCl)NPs/diatom biosilica composite.

**Table 1 biomimetics-09-00005-t001:** The molar and mass fraction of silver, chlorine, silver chloride nanoparticles, and metallic silver nanoparticles in the (Ag-AgCl)NPs/diatom biosilica composite.

Samples	Spectrum	Ag		Cl		AgClNPs		AgNPs		AgCl: AgNPs
		Mass %	n, Mole	Mass %	n, Mole	n, Mole	%	n, Mole	%	%
4.61% Ag/biosilica	46,768	4.61	0.0426	1.30	0.0366	0.0366	86	0.006	14	86:14
	46,766	5.20	0.0481	1.48	0.0416	0.0416	87	0.0065	13	87:13
	46,767	6.13	0.0567	1.65	0.0464	0.0464	82	0.0103	18	82:18
±SD		5.31 ± 0.76		1.47 ± 0.17			85 ± 3		15 ± 3	
8.49% Ag/biosilica	50,336	8.49	0.0786	1.42	0.040	0.040	51	0.0386	49	51:49
	50,333	8.58	0.0794	1.10	0.030	0.0309	39	0.0494	61	39:61
	50,335	8.89	0.0823	1.31	0.0369	0.0369	45	0.0454	55	45:55
	50,332	9.54	0.0883	1.50	0.0422	0.0422	48	0.0461	52	48:52
	50,334	9.86	0.0912	1.33	0.0374	0.0374	41	0.0538	59	41:59
±SD		9.07 ± 0.60		1.33 ± 0.15			45 ± 5		55 ± 5	

## Data Availability

Data are contained within the article.

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
