# Peer review of "Synthesis and Antimicrobial Activity of 3D Micro–Nanostructured Diatom Biosilica Coated by Epitaxially Growing Ag-AgCl Hybrid Nanoparticles"

_biomimetics, 2023, doi:10.3390/biomimetics9010005_

Round 1

Reviewer 1 Report (New Reviewer)

Comments and Suggestions for Authors

The paper reports on diatom biosilica modified by Ag and AgCl nanoparticles and its potential antimicrobial activity. The paper could be of interest but it requires major revisions before being considered for publication. I strongly suggest the authors to change the title avoiding the term "epitaxial" since it is used in case of ordered crystalline layers deposited by thermal evaporation or sputtering. Moreover, there is not a direct proof in the results presented that demonstrate the ordered stratification of AgCl and Ag on the surface of diatoms. The experimental evidence by TEM and XRD can be interpreted also as a consequence of a mixed layer of Ag nanoparticles which have been obtained by usual condensation reaction and not through ordered deposition methods. In the Introduction, the bibliography on diatom surface modified by noble metal nanoparticles could be enriched by citing two paper (Marianna Pannico; Ilaria Rea; Soundarrajan Chandrasekaran; Pellegrino Musto; Nicolas H Voelcker; Luca De Stefano, “Electroless gold-modified diatoms as surface-enhanced Raman scattering supports”, Nanoscale Research Letters 2016, 11 (1), 1-6 DOI: 10.1186/s11671-016-1539-x.; J1.   Ilaria Rea; Monica Terracciano; Soundarrajan Chandrasekaran; Nicholas Voelcker; Principia Dardano; Nicola M. Martucci; Annalisa Lamberti; Luca De Stefano, “Bioengineered silicon diatoms: adding photonic features to a nanostructured semiconductive material for biomolecular sensing” Nanoscale Research Letters 2016 11:405 DOI: 10.1186/s11671-016-1624-1.) 

All the quantities measured or esteemed in the presented work must be reported in the paper together with their errors, otherwise they are meaningless. Please, add errors in Table 1 and everywhere they are needed. Spectra reported in Figure 2 A must be correlated to the SEM images that have been used to collected them. Please, change the figure adding the SEM images where the EDX spectra have been collected. Figure 3 C shows the horizontal variation of Ag and Cl concentration and not a vertical one, so that it is not a proof of the stratification sketched in the same panel. 

The antibacterial activity of the different materials must be demonstrated by showing the evolution of the bacterial colture incubated at different time with the materials. Figure 6 does not report results, it is only a sketch summarizing the results obtained but no data are visible in the paper about that. Please, add quantitative or qualitative data of the antimicrobial activity. The experiments should include some control test, i.e. the activity of the AgCl/Ag/biosilica must be compared with those of the single materials. The utilization of diatom biosilica should be justified by a greater antibacterial efficacy otherwise it is not understandable why the authors considered their use.

Comments on the Quality of English Language

The English style and form need to be improved.

Author Response

Response to Reviewer 1

  1. Ref. No.:biomimetics- 2742565 

Synthesis and antimicrobial activity of 3D micro-nanostructured diatom biosilica coated by epitaxially growing Ag-AgCl hybrid nanoparticles.

We are grateful for your comments and insightful comments from reviewers. We have carefully reviewed the comments and edited the manuscript accordingly.  Changes to the manuscript are highlighted in red. We would also like to thank the editors and reviewers for their significant contributions to improving the manuscript

Sincerely, Myroslav Sprynskyy

Response to Reviewer #1: The paper reports on diatom biosilica modified by Ag and AgCl nanoparticles and its potential antimicrobial activity. The paper could be of interest but it requires major revisions before being considered for publication. I strongly suggest the authors to change the title avoiding the term "epitaxial" since it is used in case of ordered crystalline layers deposited by thermal evaporation or sputtering. Moreover, there is not a direct proof in the results presented that demonstrate the ordered stratification of AgCl and Ag on the surface of diatoms. The experimental evidence by TEM and XRD can be interpreted also as a consequence of a mixed layer of Ag nanoparticles which have been obtained by usual condensation reaction and not through ordered deposition methods. In the Introduction, the bibliography on diatom surface modified by noble metal nanoparticles could be enriched by citing two paper (Marianna Pannico; Ilaria Rea; Soundarrajan Chandrasekaran; Pellegrino Musto; Nicolas H Voelcker; Luca De Stefano, “Electroless gold-modified diatoms as surface-enhanced Raman scattering supports”, Nanoscale Research Letters 2016, 11 (1), 1-6 DOI: 10.1186/s11671-016-1539-x.; J1.   Ilaria Rea; Monica Terracciano; Soundarrajan Chandrasekaran; Nicholas Voelcker; Principia Dardano; Nicola M. Martucci; Annalisa Lamberti; Luca De Stefano, “Bioengineered silicon diatoms: adding photonic features to a nanostructured semiconductive material for biomolecular sensing” Nanoscale Research Letters 2016 11:405 DOI: 10.1186/s11671-016-1624-1.)

All the quantities measured or esteemed in the presented work must be reported in the paper together with their errors, otherwise they are meaningless. Please, add errors in Table 1 and everywhere they are needed. Spectra reported in Figure 2 A must be correlated to the SEM images that have been used to collected them. Please, change the figure adding the SEM images where the EDX spectra have been collected. Figure 3 C shows the horizontal variation of Ag and Cl concentration and not a vertical one, so that it is not a proof of the stratification sketched in the same panel.

The antibacterial activity of the different materials must be demonstrated by showing the evolution of the bacterial colture incubated at different time with the materials. Figure 6 does not report results, it is only a sketch summarizing the results obtained but no data are visible in the paper about that. Please, add quantitative or qualitative data of the antimicrobial activity. The experiments should include some control test, i.e. the activity of the AgCl/Ag/biosilica must be compared with those of the single materials. The utilization of diatom biosilica should be justified by a greater antibacterial efficacy otherwise it is not understandable why the authors considered their use.

We are very grateful for your critical comments and thoughtful suggestions. Based on these comments and suggestions, we have made a careful revision of the original manuscript. A revised manuscript has been submitted, in which the modified sections are highlighted in red.

Point 1: The paper reports on diatom biosilica modified by Ag and AgCl nanoparticles and its potential antimicrobial activity. The paper could be of interest but it requires major revisions before being considered for publication. I strongly suggest the authors to change the title avoiding the term "epitaxial" since it is used in case of ordered crystalline layers deposited by thermal evaporation or sputtering. Moreover, there is not a direct proof in the results presented that demonstrate the ordered stratification of AgCl and Ag on the surface of diatoms. The experimental evidence by TEM and XRD can be interpreted also as a consequence of a mixed layer of Ag nanoparticles which have been obtained by usual condensation reaction and not through ordered deposition methods.

Response 1: Our hypothetical model of the epitaxially growing Ag-AgCl hybrid nanoparticles is based on some clear evidence:

  1. Direct and unambiguous evidence of the presence of metallic silver nanoparticles and silver chloride nanoparticles in the composite are the results of X-ray diffraction analysis.
  2. The results of STEM-EDXcross-sectional profiling across two hybrid silver nanoparticles deposited on a biosilica surface clearly indicate that silver and chlorides are the main elements of these hybrid nanoparticles (the high-energy electron beam can generate X-ray at depth into the sample up to 3 μm).
  3. Stoichiometric calculations allow the estimation of the ratio of silver chlorides and metallic silver in hybrid nanoparticles (in this case 1/3).
  4. Silver chloride nanoparticles are the first to createon the biosilica surface as a result of the double displacement reaction of silver nitrate with sodium chloride, because silver chloride has very low solubility. The metallic silver nanoparticles then grow epitaxially on the surface of the silver chloride nanoparticles. The content of AgClNPs in the hybrid Ag-AgClNPs composite can be adjusted by the content of sodium chloride in the biosilica
  5. The AgNPs epitaxial grows on AgClNPs may be due to the very high surface free energy of AgClNPs and striving of AgClNPs to compensate this energy. In this case, the Ag nuclei - AgClNPs interactions are stronger than Ag nuclei -biosilica surface. So, the term "epitaxy" seems to be used appropriately in our case.

According to Report of the International Mineralogical Association (IMA) - International Union of Crystallography (IUCr) Joint Committee on Nomenclature  "Epitaxy is the phenomenon of mutual orientation of two crystals of different species, with two-dimensional lattice control (mesh in common), usually, though not necessarily, resulting in an overgrowth”. .

According to IUPAC Recommendations “Epitaxy is deposition of a crystalline over layer on a crystalline substrate” and “Epitaxial crystallization - growth of crystals on other crystals involving a precisely defined mutual orientation of their lattices. The process involves heterogeneous nucleation of the growing crystals by the substrate crystal faces”.

Point 2: In the Introduction, the bibliography on diatom surface modified by noble metal nanoparticles could be enriched by citing two paper (Marianna Pannico; Ilaria Rea; Soundarrajan Chandrasekaran; Pellegrino Musto; Nicolas H Voelcker; Luca De Stefano, “Electroless gold-modified diatoms as surface-enhanced Raman scattering supports”, Nanoscale Research Letters 2016, 11 (1), 1-6 DOI: 10.1186/s11671-016-1539-x.; J1.   Ilaria Rea; Monica Terracciano; Soundarrajan Chandrasekaran; Nicholas Voelcker; Principia Dardano; Nicola M. Martucci; Annalisa Lamberti; Luca De Stefano, “Bioengineered silicon diatoms: adding photonic features to a nanostructured semiconductive material for biomolecular sensing” Nanoscale Research Letters 2016 11:405 DOI: 10.1186/s11671-016-1624-1.).

Response 2: The suggested publications were referred in the manuscript.

Recently, diatom biosilica has been used to synthesize new 3D micro-structured composite materials. The basic 3D structure of diatom bio-silica, which plays the role of a substrate in gold modification, can as Surface-Enhanced Raman Scattering Supports [1]. The authors [2] synthesized the three-dimensional (3-D) diatom@TiO2@MnO2 composites for supercapacitor or by metabolic doping of titanium on diatomaceous biosilica to improve optical properties [3]. Diatom biosilica's composition and structure can be altered, which could increase the number of potential applications by introducing new functions and active components The diversity of the diatom silica exoskeletons morphology (over 100,000 different species) gives scientists the opportunity to choose the diatoms species that can be easily modified and correspond to the desired application, for example, in biomedicine [4,5], drug delivery system for anticancer therapeutics [6–9], for bone tissue engineering [10], as catalyst [8] and photocatalysts [11] also suitable for semiconductor nanolithography and spectroscopy, photoluminescence [12,13], in bio-triboelectric generators [14], biosensing application [15,16].

Point 3: All the quantities measured or esteemed in the presented work must be reported in the paper together with their errors, otherwise they are meaningless. Please, add errors in Table 1 and everywhere they are needed. Spectra reported in Figure 2 A must be correlated to the SEM images that have been used to collected them. Please, change the figure adding the SEM images where the EDX spectra have been collected. Figure 3 C shows the horizontal variation of Ag and Cl concentration and not a vertical one, so that it is not a proof of the stratification sketched in the same panel.

Response 3: We added errors in Table 1 in the Figure 2A modified with SEM image and mapping of raw diatom biosilica and hybrid composite.

Table 1. The molar and mass fraction of silver, chlorine, silver chloride nanoparticles and metallic silver nanoparticles in the (Ag-AgCl)NPs/diatom biosilica composite

Samples

Spectrum

Ag

Cl

AgClNPs

AgNPs

AgCl:

AgNPs

Mass %

n, mole

Mass %

n, mole

n, mole

%

n, mole

%

%

4.61 %

Ag/biosilica

46768

4.61

0.0426

1.30

0.0366

0.0366

86

0.006

14

86:14

46766

5.20

0.0481

1.48

0.0416

0.0416

87

0.0065

13

87:13

46767

6.13

0.0567

1.65

0.0464

0.0464

82

0.0103

18

82:18

±SD

5.31±0.76

1.47±0.17

   85±2.64

15±2.64

8.49 %

Ag/biosilica

50336

8.49

0.0786

1.42

0.040

0.040

51

0.0386

49

51:49

50333

8.58

0.0794

1.10

0.030

0.0309

39

0.0494

61

39:61

50335

8.89

0.0823

1.31

0.0369

0.0369

45

0.0454

55

45:55

50332

9.54

0.0883

1.50

0.0422

0.0422

48

0.0461

52

48:52

50334

9.86

0.0912

1.33

0.0374

0.0374

41

0.0538

59

41:59

±SD

9.07±0.60

1.33±0.15

44.8±4.91

55.2±4.91

SEM-EDX spectra of natural diatom biosilica and (Ag-AgCl)NPs/diatom biosilica composites are presented in Figure 2. The Figure 2 shows the SEM image of raw diatom biosilica and biosilica decorated with silver. We can observe on SEM image and SEM mapping decorated silver uniformly dispersed on the matrix of natural diatom biosilica.

Figure 2. SEM EDX spectra, SEM images and mapping of diatom biosilica samples: (A) natural biosilica, (B–C) biosilica decorated with silver (4.61 and 8.49 %, respectively).

Figure 3 C (now fig. 4c) shows the STEM-EDX the cross-sectional profile but not shows the horizontal variation of Ag and Cl concentration. The high-energy electrons generates X-ray at consid depth into the sample up to 2 μm.

Point 4: The antibacterial activity of the different materials must be demonstrated by showing the evolution of the bacterial colture incubated at different time with the materials. Figure 6 does not report results, it is only a sketch summarizing the results obtained but no data are visible in the paper about that. Please, add quantitative or qualitative data of the antimicrobial activity. The experiments should include some control test, i.e. the activity of the AgCl/Ag/biosilica must be compared with those of the single materials. The utilization of diatom biosilica should be justified by a greater antibacterial efficacy otherwise it is not understandable why the authors considered their use.

Response 4:  We added the results of antibacterial activity in the Figure.

Figure 6. Minimal inhibitory concentrations (MIC) and antimicrobial activity of obtained hybrid nanocomposite (AgCl,Ag)NPs/ diatomaceous biosilica

Our aim was to create composites supported by biocompatible diatom biosilica, compensating for the cytotoxicity of silver nanoparticles, and in order to prevent the aggregation of nanoparticles and to prevent the release of the nanoparticles into the environment. Additionally, we were interested in obtaining a synergistic antibacterial effect of silver nanoparticles.

Reviewer 2 Report (New Reviewer)

Comments and Suggestions for Authors

Review Comments

 Synthesis and antimicrobial activity of 3D micro-nanostructured diatom biosilica coated by epitaxially growing Ag-AgCl hybrid nanoparticle

 Zhanar Bekissanova, Viorica Railean, Izabela Wojtczak, Weronika Brzozowska, Alyiya Ospanova, Grzegorz Trykowski, Myroslav Sprynskyy *

Recommendation: Major Revision

The manuscript by Zhanar et al titled “Synthesis and antimicrobial activity of 3D micro-nanostructured diatom biosilica coated by epitaxially growing Ag-AgCl hybrid nanoparticle” reports synthesis of 3D micro-nanostructured diatom biosilica coated by epitaxially growing Ag-AgCl hybrid nanoparticle and their antimicrobial properties. However, this paper requires a major revision before it can be accepted for publication. The discussion on characterization results should be improved. The paper should be revised by providing a detailed discussion of the results of characterization and antimicrobial study.

The rationale of this study should be included accordingly.

The manuscript can be published after a major revision.

The following corrections should be made:

1.      Line 35-36: Hybrid nanoparticles are evenly dispersed with a dominant size of 5 to 25 in composite with an amount of 8.4 mg/g of silver. The authors should state the units of the size of the nanoparticles. Furthermore, average size distribution obtained from SEM and TEM should be stated in the abstract section.

2.      Line 39- 42: The nanocomposites containing (Ag-AgCl)NPs and natural diatom biosilica showed resistance to bacterial strains from the American Type Cultures Collection and Clinical Isolates (Diabetic Foot Infection and Wound  Isolates). The authors should provide quantified results from their experiments in order to justify the antibacterial resistance claim. It has been proved that AgCl on its own can combat bacteria. Can the authors provide control experimental results involving using Ag+? This is significant in order to support this study and compare the results with the synthesized nanocomposites.

3.      Line 57- 59:  Diatom biosilica is recently actively used to synthesize new 3D micro-nanostructured composite materials. The statement should read: Recently, Diatom biosilica has been used to synthesize new 3D micro-structured composite materials. The authors should provide references to showcase the reports in which diatom biosilica has been recently used to synthesize new 3D micro-nanostructured composite materials.

4.      Line 86- 87: The Silver chloride nanoparticles (AgClNPs) can be obtained by ion-exchange reaction by introducing salts NaCl [22,25] or AlCl3•6H2O [26]. The authors should correct the chemical formulae of AlCl3•6H2O to AlCl3•6H2O.

5.      Line 87-89: There are few reports on the synthesis of hybrid silver nanoparticles combining metallic silver and silver chloride nanoparticles. The authors should clearly indicate the rationale of synthesis of hybrid silver nanoparticles combining metallic silver and silver chloride nanoparticles. Furthermore, the rationale of this study should be stated accordingly.

6.      AgNO3 should be corrected to AgNO3 though out the entire manuscript.

7.      Line 159: AgNO₃(aq) + NaCl(s) = NaNO₃(aq) + AgCl(s) (1). Authors should replace = with an   arrow (         ). This should apply to equation 2 and 3 too. The products should be shown using an arrow sign and not the = sign. The authors must do the corrections in the entire manuscript to conform with norms of writing chemical equations.

8.       Line 217: The results of FTIR spectroscopy analysis [30,31] showed the presence…. Can authors provide the figure showing the FTIR Image?

9.      Line 281-284:  In TEM micrographs (Ag-AgCl) NPs hybrid nanoparticles can be observed as dark and light shades. Particles with darker shades are AgNPs, while lighter shades may be AgClNPs. The authors should justify the statement” Particles with darker shades are AgNPs, while lighter shades may be AgClNPs”. Can the authors provide a proof to justify the statement?

10.  The TEM images of the 3D micro-nanostructured hybrid (Ag-AgCl) NPs/diatom biosilica composite (Figure 3) show spherical and quasi-spherical morphologies. Can authors explain why they did not synthesize uniform shape?

11.  The XRD diffraction peaks reveal the predominant peaks of AgCl in comparison to Ag. Do the authors have XPS data validate the formation of AgCl and Ag nanocomposites? XPS data should be provided.

12.  Authors should provide XPS data for this work in order to validate formation of 3D micro-nanostructured diatom biosilica coated by epitaxially growing Ag-AgCl hybrid nanoparticle.

13.  The authors should provide a detailed discussion on XRD in order to distinguish the XRD patterns of AgCl and Ag nanocomposites.

14.  The authors should include a comparison of the antimicrobial activities of their synthesized 3D micro-nanostructured diatom biosilica coated by epitaxially growing Ag-AgCl hybrid nanoparticle with those in literature. Why was this study conducted? Kindly explain the rationale of your study. Is the synthesized 3D micro-nanostructured diatom biosilica coated by epitaxially growing Ag-AgCl hybrid nanoparticle better than the reported antimicrobial activities?

15.  Furthermore, this paper lacks detailed discussion of the results especially the characterization techniques. Authors should revise this manuscript by providing candid discussion of the results, FTIR, XRD, and antimicrobial studies.

Comments on the Quality of English Language

Editing of English required

Author Response

Response to Reviewer 2

  1. Ref. No.:biomimetics- 2742565  

Synthesis and antimicrobial activity of 3D micro-nanostructured diatom biosilica coated by epitaxially growing Ag-AgCl hybrid nanoparticles.

We are grateful for your comments and insightful comments from reviewers. We have carefully reviewed the comments and edited the manuscript accordingly. Changes to the manuscript are highlighted in red. We would also like to thank the editors and reviewers for their significant contributions to improving the manuscript

Sincerely, Myroslav Sprynskyy

Response to Reviewer #2: The manuscript by Zhanar et al titled “Synthesis and antimicrobial activity of 3D micro-nanostructured diatom biosilica coated by epitaxially growing Ag-AgCl hybrid nanoparticle” reports synthesis of 3D micro-nanostructured diatom biosilica coated by epitaxially growing Ag-AgCl hybrid nanoparticle and their antimicrobial properties. However, this paper requires a major revision before it can be accepted for publication. The discussion on characterization results should be improved. The paper should be revised by providing a detailed discussion of the results of characterization and antimicrobial study.

The rationale of this study should be included accordingly.

The manuscript can be published after a major revision.

The following corrections should be made:

We are very grateful for your critical comments and thoughtful suggestions. Based on these comments and suggestions, we have made a careful revision of the original manuscript. A revised manuscript has been submitted, in which the modified sections are highlighted in red.

Point 1:  Line 35-36: Hybrid nanoparticles are evenly dispersed with a dominant size of 5 to 25 in composite with an amount of 8.4 mg/g of silver. The authors should state the units of the size of the nanoparticles. Furthermore, average size distribution obtained from SEM and TEM should be stated in the abstract section.

Response 1: The units and average size of nanoparticles are added. Hybrid nanoparticles are evenly dispersed with a dominant size of 5 to 25 nm in composite with an amount of 8.4 mg/g of silver. The average size of nanoparticles is 7.5 nm also there are nanoparticles with a size of 1-2 nm and particles of 20-40 nm.

Point 2:  Line 39- 42: The nanocomposites containing (Ag-AgCl)NPs and natural diatom biosilica showed resistance to bacterial strains from the American Type Cultures Collection and Clinical Isolates (Diabetic Foot Infection and Wound  Isolates). The authors should provide quantified results from their experiments in order to justify the antibacterial resistance claim. It has been proved that AgCl on its own can combat bacteria. Can the authors provide control experimental results involving using Ag+? This is significant in order to support this study and compare the results with the synthesized nanocomposites.

Response 2: The quantified results of antibacterial activity is added in the Figure.

Point 3: Line 57- 59:  Diatom biosilica is recently actively used to synthesize new 3D micro-nanostructured composite materials. The statement should read: Recently, Diatom biosilica has been used to synthesize new 3D micro-structured composite materials. The authors should provide references to showcase the reports in which diatom biosilica has been recently used to synthesize new 3D micro-nanostructured composite materials.

Response 3: Recently, diatom biosilica has been used to synthesize new 3D micro-structured composite materials. The basic 3D structure of diatom bio-silica, which plays the role of a substrate in gold modification, can as Surface-Enhanced Raman Scattering Supports [1]. The authors [2] synthesized the three-dimensional (3-D) diatom@TiO2@MnO2 composites for supercapacitor or by metabolic doping of titanium on diatomaceous biosilica to improve optical properties [3].

Point 4:  Line 86- 87: The Silver chloride nanoparticles (AgClNPs) can be obtained by ion-exchange reaction by introducing salts NaCl [22,25] or AlCl3•6H2O [26]. The authors should correct the chemical formulae of AlCl3•6H2O to AlCl3•6H2O.

Response 4: The chemical formulae AlCl3•6H2O corrected to AlCl3•6H2O.

Point 5: Line 87-89: There are few reports on the synthesis of hybrid silver nanoparticles combining metallic silver and silver chloride nanoparticles. The authors should clearly indicate the rationale of synthesis of hybrid silver nanoparticles combining metallic silver and silver chloride nanoparticles. Furthermore, the rationale of this study should be stated accordingly.

Response 5: The information about the synthesis of hybrid silver nanoparticles combining metallic silver and silver chloride nanoparticles is added to the manuscript. AgCl-Ag hybrid particles have less cytotoxic effect compared to pure AgCl nanoparticles [32], low cytotoxicity was also detected in alginate-based films containing Ag/AgCl nanoparticles [33]. Potential antibacterial agents of the composition MOFs of Cu(I)bpyCl (bpy = 4,4′-bipyridine) formulated with AgCl/Ag nanoparticles with possible synergistic effects were synthesized [34]. The composites based on AgCl/Ag nanoparticles coated on reduced graphene oxide have antimicrobial potential and the action of healing burn wounds [35]. In the work AgCl/Ag nanoparticles decorated layered double hydroxide, in which layered double hydroxide plays the role of a support and prevents aggregation of nanoparticles [36].

Point 6: AgNO3 should be corrected to AgNO3 though out the entire manuscript.

Response 6:  We corrected the chemical formulae AgNO3 to AgNO3.

Point 7: Line 159: AgNO₃(aq) + NaCl(s) = NaNO₃(aq) + AgCl(s) (1). Authors should replace = with an   arrow (         ). This should apply to equation 2 and 3 too. The products should be shown using an arrow sign and not the = sign. The authors must do the corrections in the entire manuscript to conform with norms of writing chemical equations.

Response 7: It was done.

Point 8: Line 217: The results of FTIR spectroscopy analysis [30,31] showed the presence…. Can authors provide the figure showing the FTIR Image?

Response 8: Unfortunately, we did not perform infrared spectroscopy analysis of the synthesized composites. The FTIR spectra of the used diatom biosilica have already been published in earlier articles. In this work we only reported the quoted FTIR data from these previous works.

Point 9: .      Line 281-284:  In TEM micrographs (Ag-AgCl) NPs hybrid nanoparticles can be observed as dark and light shades. Particles with darker shades are AgNPs, while lighter shades may be AgClNPs. The authors should justify the statement” Particles with darker shades are AgNPs, while lighter shades may be AgClNPs”. Can the authors provide a proof to justify the statement?

Response 9: Metallic silver nanoparticles and silver chloride nanoparticles differ in crystal structure and density, which can create this contrast. But you are right, we do not have confirmation of this statement. This could be confirmed by more detailed studies using the STEM-EDX profiling method. Unfortunately, this was not possible due to limited access to TEM  technique.

Point 10: The TEM images of the 3D micro-nanostructured hybrid (Ag-AgCl) NPs/diatom biosilica composite (Figure 3) show spherical and quasi-spherical morphologies. Can authors explain why they did not synthesize uniform shape?

Response 10: Different factors (e.g. time, concentration, pH, temperature, pressure, structure, surface etc.) can influence the sizes and shapes of the synthesized nanoparticles. In our case, the synthesis does not take place in a perfectly homogeneous environment, and hybrid nanoparticles consisting of two nanoparticles with different crystal structures are synthesized. So it is difficult to expect a perfect uniform shape of hybrid nanostructures.

Point 11: The XRD diffraction peaks reveal the predominant peaks of AgCl in comparison to Ag. Do the authors have XPS data validate the formation of AgCl and Ag nanocomposites? XPS data should be provided.

Response 11: The X-ray diffraction is a fingerprint method in the indentyfication of the crystalline phases in a mixture. Experimental XRD patterns were compared to reference patterns (AgNPs - JCPDS N 04-0783; AgCl - JCPDS N 31-1238) to determine what crystalline form of silver nanoparicles are present in syntesyed composites. The position of the reference peaks were very good match.  The use of the STEM-EDX technique allowed for the determination of the elementary composition of an individual nanoparticles throughout its entire cross-section (dept up to 1000 nm and more). X-ray Photoelectron Spectroscopy (XPS) is alloved for the analyses the chemical bonding state of the sample surface (dept up to 10 nm). Unfortunately, we do not have access to this technique. Furthermore, the XPS method does not provide the option of the analyses of individual nanoparticle.

Point 12: Authors should provide XPS data for this work in order to validate formation of 3D micro-nanostructured diatom biosilica coated by epitaxially growing Ag-AgCl hybrid nanoparticle.

Response 12:  Unfortunately, we do not have XPS to perform such an analysis, but the data obtained from this analysis would undoubtedly be very interesting. 

Direct and unambiguous evidence of the presence of metallic silver nanoparticles and silver chloride nanoparticles in the composite are the results of X-ray diffraction analysis. The STEM-EDX profiling results also provide convincing evidence of the presence of hybrid nanoparticles on the biosilica surface in the form of epitaxial intergrowths of metallic silver nanoparticles and silver chloride nanoparticles.

Point 13: The authors should provide a detailed discussion on XRD in order to distinguish the XRD patterns of AgCl and Ag nanocomposites.

Response 13: The X-ray diffraction is a fingerprint method in the indentyfication of the crystalline phases in a mixture. Experimental XRD patterns were compared to reference patterns (AgNPs - JCPDS N 04-0783; AgCl - JCPDS N 31-1238) to determine what crystalline form of silver nanoparicles are present in syntesyed composites. Identification of the crystalline form of nanoparticles is based on XRD reference spectra. Since the composites are synthesized for the first time, broader discussion is limited.

Point 14: The authors should include a comparison of the antimicrobial activities of their synthesized 3D micro-nanostructured diatom biosilica coated by epitaxially growing Ag-AgCl hybrid nanoparticle with those in literature. Why was this study conducted? Kindly explain the rationale of your study. Is the synthesized 3D micro-nanostructured diatom biosilica coated by epitaxially growing Ag-AgCl hybrid nanoparticle better than the reported antimicrobial activities?

Response 14: This information is incorporated into the manuscript.

Natural matrices have the necessary properties for the synthesis of biocompatible non-toxic composite materials [37,38]. When synthesizing composite materials based on natural matrices, they prevent emission of silver nanoparticles in environment and aggregation of silver nanoparticles [36].

The aim of the research is to synthesize the new 3D nano-microstructured composite with antimicrobial activity based on diatom biosilica coated by epitaxially growing Ag-AgCl hybrid nanoparticles. The antimicrobial activity was investigated against Gram-positive bacteria Staphylococcus aureus (from American Type Culture Collection and Diabetic Foot Infection Isolate), Gram-negative bacteria Klebsiella pneumoniae and Escherichia coli strains (from American Type Culture Collection and Wound Isolate).

The present study is the first to produce AgCl-Ag hybrid nanoparticles based on nanostructured diatomaceous biosilicon. Antimicrobial potential against diabetic foot isolates and wound isolates (DFII and WI) was studied to investigate antibacterial activity in addition to ATCC strains.

Nanomaterial Ag/AgCl/rGO showed minimum inhibitory concentration (MIC) against E. coli and S. aureus were 2 and 4 mg L-1, respectively (in terms of the Ag element) [35]. The antibacterial activity of composite with AgCl/Ag nanoparticles against E. coli with MIC of ∼7.8 μg mL–1, while the MIC value was ∼16 μg mL–1 against S. aureus [34].

Point 15: Furthermore, this paper lacks detailed discussion of the results especially the characterization techniques. Authors should revise this manuscript by providing candid discussion of the results, FTIR, XRD, and antimicrobial studies.

Response 15: Discussion on the synthesis of hybrid silver nanoparticles combining metallic silver and silver chloride nanoparticles was added to the manuscript, as well as a discussion on the results of antimicrobial tests. Information about the research techniques used in this work along with the research conditions used is presented in the Methods and Materials section. We also want to point out that 3D micro-nanostructured diatom biosilica coated by epitaxially growing Ag-AgCl hybrid nanoparticles is a material synthesized for the first time. Therefore, a broader comparative discussion of the obtained results (XRD, SEM, TEM, PL, zeta) is also somewhat limited.

Round 2

Reviewer 1 Report (New Reviewer)

Comments and Suggestions for Authors

Some errors should be corrected since they must be written with only one significant digit (at least two, if the first is 1).  This means that 44.8±4.91 must be 45±5. Please, check all the errors format in tables and in text.

Comments on the Quality of English Language

English style and form are good.

Author Response

Response to Reviewers Comments

  1. Ref. No.:biomimetics-2742565  

Synthesis and antimicrobial activity of 3D micro-nanostructured diatom biosilica coated by epitaxially growing Ag-AgCl hybrid nanoparticles.

  We would like to thank the editors and reviewers for their significant contributions to improving the manuscript

Sincerely,

Response to Reviewer #1:

Some errors should be corrected since they must be written with only one significant digit (at least two, if the first is 1).  This means that 44.8±4.91 must be 45±5. Please, check all the errors format in tables and in text. 

Thank you very much for the remark. An appropriate correction has been made.

Reviewer 2 Report (New Reviewer)

Comments and Suggestions for Authors

The authors have addressed my concerns.

Comments on the Quality of English Language

Editing of English is required.

The manuscript can be accepted for publication after moderate editing of English.

Author Response

Response to Reviewers Comments

  1. Ref. No.:biomimetics-2742565  

Synthesis and antimicrobial activity of 3D micro-nanostructured diatom biosilica coated by epitaxially growing Ag-AgCl hybrid nanoparticles.

  We would like to thank the editors and reviewers for their significant contributions to improving the manuscript

Sincerely,

Response to Reviewer #2: The authors have addressed my concerns. Editing of English is required.

Authors appreciate the reviewer's attention. We have tried to improve the quality of the manuscript. The final text has been checked and revised to improve understanding of the most complex sentences.

This manuscript is a resubmission of an earlier submission. The following is a list of the peer review reports and author responses from that submission.

Round 1

Reviewer 1 Report

Comments and Suggestions for Authors

Comments

The authors prepared Ag/AgCl NPs deposited 3D micro-nanostructured diatom biosilicas and investigated their antimicrobial activity. However, the experiments presented in this manuscript is not well designed. And the characterizations are also insufficient to prove the statements in the manuscript. Overall, I believe that this manuscript is immature to be accepted in an academic journal.

1.      As well know, AgNPs have excellent antimicrobial activity, which is also proved by the antimicrobial experiment in this manuscript. So Why AgCl NPs are introduced to the composites?

2.      Since ‘3D micro-nanostructured’ is demonstrated in the title of this manuscript, the structure of the diatom biosilicas should be fully characterized.

3.      In the graphitic, the authors proposed the structure of Ag/AgCl NPs. Is there any evidence to prove the hypothesis? The elemental content analysis in Figure 2 is insufficient to prove the statement.

4.      The quantitative analysis of the percentage of Ag and Cl is incorrect by EDS, which is a semiquantitative method. ICP should be used to precisely determine the percentage of Ag and Cl, which is very important to confirm the formation of the hybrid nanoparticle.

5.      The analysis of Figure 2C and the proposed model of the hybrid nanoparticle is incorrect.

6.      In Figure 3, why dose the formation of Ag and Ag/AgCl NPs decrease the zeta potential of the composite significantly?

7.      The full characterization of the composite with Ag content of 5.49% should be presented in the manuscript.

8.      In Figure 4, the normalized PL spectra seems strange.

9.      The written of this manuscript is lack of standardization, for instance, the abbreviations, the citation of the figures etc.

Comments on the Quality of English Language

The English Language of this manuscript is fine.

Reviewer 2 Report

Comments and Suggestions for Authors

The manuscript entitled "Synthesis and antimicrobial activity of 3D micro-nanostructured diatom biosilica coated by epitaxially growing Ag-AgCl 3 hybrid nanoparticles"  is clear, well-written and reports the preparation and characterization of Ag nanoparticles in biosilica.

I want  to comment on some topics: Concerning the PL spectroscopy, I suggest some questions to improve the discussion. The authors describe, in the text, the results observed in the graphics. However, the intensity of the spectra is not mentioned. the question is: why does the silver nanoparticle show quenching of the Biosilica PL? This effect was observed also in the author´s ulterior reports when using biosilica doped with Ag/Ce nanoparticles. 

Another comment is concerned with the MIC and Antimicrobial effect.  Why different concentration, 4,61% and 8,49%, displayed the same results for antimicrobial effect?  In fact as the MIC tested in solution of Ag/AgCl/ Biosilica  it supposes an chemical equilibrium of Ag-AgCl.   Could this mask the effective result of antimicrobial activity?